# Chromoanagenesis Landscape in 10,000 TCGA Patients

**DOI:** 10.3390/cancers13164197

**Published:** 2021-08-20

**Authors:** Roni Rasnic, Michal Linial

**Affiliations:** 1The Rachel and Selim Benin School of Computer Science and Engineering, The Hebrew University of Jerusalem, Jerusalem 9190401, Israel; 2Department of Biological Chemistry, Institute of Life Sciences, The Hebrew University of Jerusalem, Jerusalem 9190401, Israel; michall@mail.huji.ac.il

**Keywords:** CNA, cancer, TP53, chromothripsis, mutual exclusivity

## Abstract

**Simple Summary:**

Chromoanagenesis is a single catastrophic event in which one or few chromosomes are shattered and disorderly reassembled. Chromoanagenesis is common in many types of cancers. In this study, we utilize data from The Pan-Cancer Analysis of Whole Genome (PCAWG) to build a machine learning algorithm that detects chromoanagenesis with high accuracy. We applied the algorithm on ~10,000 samples from The Cancer Genome Atlas (TCGA), thereby providing, for the first time, chromoanagenesis status labels for the complete data set. An in-depth analysis of somatic and clinical chromoanagenesis features is presented for 20 cancer types. Mutual exclusivity patterns between genes impaired in chromoanagenesis versus non-chromoanagenesis cases might imply at distinct pathways involved in chromoanagenesis-driven tumorigenesis.

**Abstract:**

During the past decade, whole-genome sequencing of tumor biopsies and individuals with congenital disorders highlighted the phenomenon of chromoanagenesis, a single chaotic event of chromosomal rearrangement. Chromoanagenesis was shown to be frequent in many types of cancers, to occur in early stages of cancer development, and significantly impact the tumor’s nature. However, an in-depth, cancer-type dependent analysis has been somewhat incomplete due to the shortage in whole genome sequencing of cancerous samples. In this study, we extracted data from The Pan-Cancer Analysis of Whole Genome (PCAWG) and The Cancer Genome Atlas (TCGA) to construct and test a machine learning algorithm that can detect chromoanagenesis with high accuracy (86%). The algorithm was applied to ~10,000 unlabeled TCGA cancer patients. We utilize the chromoanagenesis assignment results, to analyze cancer-type specific chromoanagenesis characteristics in 20 TCGA cancer types. Our results unveil prominent genes affected in either chromoanagenesis or non-chromoanagenesis tumorigenesis. The analysis reveals a mutual exclusivity relationship between the genes impaired in chromoanagenesis versus non-chromoanagenesis cases. We offer the discovered characteristics as possible targets for cancer diagnostic and therapeutic purposes.

## 1. Introduction

Over the past decade, the term chromoanagenesis (for chromosome rebirth) was coined to describe a catastrophic cellular event in which large numbers of complex rearrangements occur at one or a few chromosomal loci. A chromoanagenesis event consists of multiple chromosomal breakage and results in a variety of chromosomal abnormalities, including copy number alterations (CNA), inversions, and inter-and intra-chromosomal translocations. Chromoanagenesis was originally discovered in tumor cells and in individuals with congenital disorders [1]. It was also found in healthy individuals [2]. The full extent and impact of the different types of chromoanagenesis remain unknown. Most commonly, whole genome sequencing is applied in order to identify the phenomenon.

There are three subtypes of chromoanagenesis: chromothripsis, chromoplexy, and chromoanasynthesis [3,4]. Chromothripsis is the localized shattering and reshuffling of tens to hundreds of chromosome segments. Micronuclei formation is considered to be the source of chromothripsis [5], and the non-homologous end joining is presumed to be chromothripsis reassembly measure [1]. Chromoplexy is characterized by the interdependent occurrence of multiple inter- and intra-translocations and deletions resulting from double stranded breaks with precise junctions [6]. Non-homologous end joining is also presumed to be a chromoplexy main reassembly measure. Chromoanasynthesis is a replication-based complex rearrangement process. The breakpoint junctions in chromoanasynthesis show micro-homology and template insertions, consistent with defective DNA replications and suggesting the involvement of error-prone DNA replication pathways [6].

The most exhaustive research on chromoanagenesis was performed as a part of The Pan-Cancer Analysis of Whole Genomes (PCAWG) study [7]. PCAWG analysis included 2658 cancer genomes and their matching normal tissues across 38 tumor types. The PCAWG study confirmed that chromoanagenesis is common in many cancer types. There is an overlap of 799 samples (from 22 tumor types) between PCAWG and The Cancer Genome Atlas (TCGA) project. We utilized this overlap to curate a data set with TCGA genic CNA data, and PCAWG chromoanagenesis labeling. We utilized the curated data set to create a highly accurate machine learning model that identifies chromoanagenesis and employed it on additional ~10,000 cancerous samples from TCGA.

In this study we performed an in-depth analysis of chromoanagenesis’ somatic, cancer-type specific characteristics while focusing on coding genes. Many of the found somatic single nucleotide variants (SNV) and CNA patterns match previous studies, as we concur on chromoanagenesis related genes. Additionally, we identified CNA and cancer-type specific mutual-exclusivity patterns matching established observations that were reported regardless of chromoanagenesis. We offer our TCGA samples classification and novel discoveries as the basis for further investigating chromoanagenesis.

## 2. Materials and Methods

### 2.1. Study Population

Masked CNA data at the gene level for 10,728 TCGA individuals was downloaded from the GDC portal (https://portal.gdc.cancer.gov/ (accessed on 1 August 2020). The data does not include genes in the Y chromosome. The PCAWG project performed a whole genome analysis of 799 of those individuals. PCAWG thoroughly described each individual chromosomal state. The data was downloaded from the Chromothripsis Explorer site (http://compbio.med.harvard.edu/chromothripsis/ (accessed on 3 June 2020). The site includes chromothripsis labeling, as well as a labeling for other complex chromosomal events [8]. The labeling for other complex chromosomal events does not include additional details regarding the nature of the events in each individual. We reduced the description to include only whether an individual had chromothripsis, other complex chromosomal events or not. For more advanced analysis, we considered an individual with chromothripsis and/or other complex chromosomal events as having chromoanagenesis. We also extracted from TCGA masked SNV data (from the MuTect2 pipeline variant data, including variant annotation) and clinical and exposure data. HNSC HPV status was extracted from the Lawrence, M. et al. study [9].

### 2.2. Machine Learning Pipeline

We used the data of the 799 individuals examined by PCAWG as the basis for our ML model selection and training. We used 70% of the data as training samples, 15% as model and feature selection testing data (development testing data), and another 15% of the data as the final test set, only used after the model was finalized and feature selection was completed. The selected model, presenting the best results on the development testing data was sklearn’s DecisionTreeClassifier [10].

We tested multiple features designed to capture copy number oscillation patterns in the data. We considered an oscillation to be an adjacent collection of genes from the same chromosomal arm with the same CNA. The examined features included: overall number of amplifications, overall number of deletions, maximal and mean CNA length (in genes), number of CNA in highly varied chromosomes, maximal number of oscillations (in all chromosomal arms), and several features designed to reflect the relations between the maximal number of oscillations in chromosomal arm to the mean number of oscillations in all chromosomal arms. After careful consideration of the different features, we manually choose features with both relatively high correlation to chromoanagenesis status and small overlap with other chosen features.

We represented each individual with the chosen features. The optimal model used only features concerning the distribution of oscillation number in the chromosomal arms. The model selected to use only the two most informative features: (i) max number of oscillations in a chromosomal arm −3* mean number of oscillations in all chromosomal arms. (ii) standard deviation of the number of oscillations across all chromosomal arms. We assessed our models according to their accuracy rate. Accuracy is defined as [true positive + true negative]/[all positive + all negative]. This model presented the best results for the development testing data, and reached 85.7% accuracy on the final testing data.

### 2.3. Statistical Analysis

We applied Fisher’s exact test (using scipy stats module [11]) when testing differences in chromoanagenesis genic CNA. We applied the same methodology when comparing the number of per-gene somatic mutation types across chromoanagenesis and non-chromoanagenesis samples. We chose a significance threshold of $5\times 10^{-7}$; which is based on performing a Bonferroni correction for 20,000 genes, with a conservative threshold of 0.01.

### 2.4. Visualization

Matplotlib [12] and seaborn [13] were used to generate the boxplot visualization representing interquartile range (IQR) including, 25th percentile, median, 75th percentile, and 1.5 * IQR for the whiskers. Matplotlib was also used to create Figures 2 and 3 and all Manhattan and Kaplan–Meier plots.

## 3. Results

Chromoanagenesis status for 799 cancer samples from 22 cancer types in the TCGA cohort was collected via PCAWG (see Methods). Overall, 371 of the samples (46.4%) had chromoanagenesis. The chromoanagenesis samples can be further divided: 64 have chromothripsis, 143 have both chromothripsis and other complex chromosomal events, and 164 with only other, non-chromothripsis, complex chromosomal events.

Chromoanagenesis frequency varied greatly between cancer types, ranging from 3% in thyroid carcinoma (THCA) to 88% in glioblastoma (GBM). Chromoanagenesis subtype distributions also varied among cancer types. For example, 75% of kidney renal clear cell carcinoma (KIRC) chromoanagenesis samples had chromothripsis, while 57% of liver hepatocellular carcinoma (LIHC) chromoanagenesis samples had strictly non-chromothripsis events (Figure 1a).

### 3.1. Cancer Type Impacts Genic CNA Frequency

We collected masked genic CNA from TCGA for the 799 samples. Namely, for each sample and for 19,729 known coding genes, we know whether the number of copies in the somatic sample is higher, identical, or smaller than in the matching germline sample (see Methods). We chose to use genic CNA (derived from GISTIC [14] results) and not whole genome CNA to reduce noise and limit data dimensionality. Notably, genic CNA cannot capture all key chromoanagenesis features. Specifically, we cannot detect inter-chromosomal and intra-chromosomal translocations or inversions. Similarly, deletions or insertions in intergenic regions are not recorded. Accordingly, chromoplexy, which has less CNA than other types of chromoanagenesis, might be missed (Figure 1b).

We examined the total number of genes with CNA for each of the four chromoanagenesis states: (i) no chromoanagenesis; (ii) chromothripsis; (iii) chromothripsis and other complex chromosomal events; (iv) non-chromothripsis complex chromosomal events. In all three chromoanagenesis groups, the number of genes with CNA was significantly higher. The mean number of genes with altered copy number is 559.2 for the no chromoanagenesis group, 701.2 for chromothripsis, 1268.5 for samples with both chromoanagenesis and other complex events and 964.1 for non-chromothripsis complex events. One-way ANOVA test yields a *p*-value of $1.5\times 10^{-24}$ (Figure 2a). Similar results and trends were observed when examining separately CNA for deletion or amplification events (Appendix A).

The variability in the total number of CNAs is heavily influenced by cancer type, as different cancer types have wildly distinct somatic characteristics. The TCGA–PCAWG samples are distributed unevenly among cancer types and chromoanagenesis status. Therefore, the total number of genic CNA will not suffice to identify a sample’s chromoanagenesis status. For example, for BRCA (Breast invasive carcinoma) and LUSC (lung squamous cell carcinoma), a one-way ANOVA test on chromoanagenesis number of genic CNA yields non-significant *p*-values of 0.42 and 0.88, respectively (Figure 2b). The genic CNA distribution among all 33 cancer types in TCGA is presented in Figure 2c. The different cancer-type samples exhibit huge variability in the number of altered genes. The mean number of altered genes per cancer type range over 2–3 orders of magnitude with minimal number in thyroid carcinoma (THCA) and maximal in ovarian serous cystadenocarcinoma (OV).

### 3.2. Predicting Chromoanagenesis at High Accuracy

To overcome the variation in somatic background and chromoanagenesis proportions between cancer types, we examined more complex genic CNA attributes as chromoanagenesis status predictors. When examining adjacent genes on the same chromosomal arm, it is likely that a similar CNA status (i.e., amplification or deletion) is attributed to the same CNA event. The alternative possibility of having unrelated similar CNA events in adjacent genes is less probable. We used this assessment to measure different CNA features for each chromosomal arm. For example, the number of genes affected by the same CNA or the number of gene-affecting CNA per chromosomal arm.

A key indicator for chromoanagenesis is CNA oscillations along the affected chromosomes. We calculated the number and length (in genes) of oscillations per chromosomal arm. Interestingly, the chromoanagenesis samples include one or few chromosomal arms with exceptionally high number of oscillations. Contrastingly, the non-chromoanagenesis samples (with many oscillations) include many chromosomal arms with a high number of oscillations (Figure 3). This attribute is exactly what is expected in the context of chromoanagenesis. Consequently, some features were engineered to express the difference between maximal number of oscillations (in a chromosomal arm) and the average number of oscillations (across all chromosomal arms).

We trained a decision tree on 85% of the data to identify whether a sample has chromoanagenesis (see Methods). We reached an 85.7% accuracy rate, with 88.9% accuracy, 83.6% specificity, and 78.4% sensitivity for having chromoanagenesis. This generic somatic chromoanagenesis detection module is applicable to any cancer type. However, due to the limited sample size and previously discussed data limitations, we were not able to distinguish between different chromoanagenesis subtypes (i.e., chromothripsis and other chromosomal events).

The machine learning model was applied on all remaining 9929 TCGA samples from all 33 cancer types reported in TCGA (Figure 4a, Appendix A). Overall, we classified 3892 individuals (39.2%) as having chromoanagenesis. Figure 4b describes the predicted percentage of chromoanagenesis for each cancer type. We also marked 10 cancer types that were not examined by PCAWG and therefore were not a part of the model training. When comparing our results to PCAWG-verified chromoanagenesis identification (of samples from TCGA and the International Cancer Genome Consortium), we observed that the chromoanagenesis rate is very similar for most cancer types, with $R^{2}$ of 0.7461 (Figure 4c). Notably, the high correlation was evident across all cancer types, despite the limited sample size for some cancer types in PCAWG.

### 3.3. Chromoanagenesis Cancer Specific CNA Patterns

An in-depth analysis was performed to uncover cancer type specific chromoanagenesis CNA patterns. For each gene, we tested whether the frequency of CNA in each sample type (chromoanagenesis and non-chromoanagenesis) was significantly different (see Methods). We limited the analysis to the 20 cancer types with at least 50 chromoanagenesis samples and 50 non-chromoanagenesis samples: BLCA (Bladder Urothelial Carcinoma), BRCA (Breast invasive carcinoma), CESC (Cervical squamous cell carcinoma and endocervical adenocarcinoma), COAD (Colon adenocarcinoma), ESCA (Esophageal carcinoma), GBM (Glioblastoma multiforme), HNSC (Head and Neck squamous cell carcinoma), KIRC (Kidney renal clear cell carcinoma), LGG (Brain Lower Grade Glioma), LIHC (Liver hepatocellular carcinoma), LUAD (Lung adenocarcinoma), LUSC (Lung squamous cell carcinoma), OV (Ovarian serous cystadenocarcinoma), PAAD (Pancreatic adenocarcinoma), PRAD (Prostate adenocarcinoma), READ (Rectum adenocarcinoma), SARC (Sarcoma), SKCM (Skin Cutaneous Melanoma), STAD (Stomach adenocarcinoma), and UCEC (Uterine Corpus Endometrial Carcinoma).

The emerging patterns mostly consisted of numerous adjacent genes and often included more than one chromosomal region. Some CNA regions showed clear distinction between copy number deletion and amplification, while other regions were significantly altered, but not specifically enriched with either deletions or amplifications (Figure 5 and Appendix A). Unsurprisingly, the vast majority of the significantly altered chromosomal regions are associated with chromoanagenesis samples, while in the non-chromoanagenesis samples, the copy number of the chromosomal regions is maintained. Namely, the presented associations mostly indicate high CNA frequency in the chromoanagenesis samples and very low CNA frequency in the non-chromoanagenesis samples. The number of significant association regions varies greatly between cancer types; UCEC is extremely abundant in statistically significant regions for any CNA (Table 1).

Various CNA association patterns are linked to the different cancer types. Notably, amplifications are more common than deletions (Table 1). Despite differences between cancer types, some CNA regions display the same phenomenon in chromoanagenesis in several cancer types (Appendix A). For instance, both BRCA and COAD have deletions in region 8p11.21, which occur around the same set of genes, and include the known driver gene HOOK3 [15]. Additional examples include STAD and UCEC that exhibit similar deletion patterns in the 4q34–4q35 region, which includes tumor suppressor gene FAT1. The deletion of FAT1 promotes malignant progression [16]. BRCA, STAD, and UCEC have similar large amplifications in regions 17q12–17q21, previously observed in breast and gastric cancers [17,18]. This gene rich large region includes several known cancer genes. An amplification in the gene ERBB2 was shown to occur in breast cancer with a high rate of proliferation [19,20].

Three of the examined cancer types—ESCA, PAAD, and READ—did not have any statistically significant copy number alterations, in either chromosomal region or gene level. The lack of significant results is mostly explained by the relatively small number of samples for either chromoanagenesis or non-chromoanagenesis samples. Applying a more relaxed significance threshold will likely reveal additional results, for all tested cancer types. Figure 5 and Appendix A depict Manhattan plots for deleted, amplified, and altered regions for each of the 20 cancer types. Appendix A details the significantly altered chromosomal region per cancer type. The gene-level *p*-values are summarized in Appendix A.

### 3.4. Chromoanagenesis Single Gene Focal Alterations

Some prominent and significant CNA consist only of a single gene. We consider a gene as a distinct CNA gene if the Fisher’s exact test *p*-value passes the predefined significance threshold of $5\times 10^{-7}$, and is at least ×2.5 orders-of-magnitude more significant than its adjacent genes. For UCEC, we applied a threshold of × 4 orders-of-magnitude, to mitigate the extreme results in this cancer type. The analysis revealed several deleted genes: LRP1B, PDE4D, DLG2, ANKS1B, WWOX, and DMD. LRP1B (a known tumor suppressor) deletion was associated with chemotherapy resistance in high-grade cancers [21]. The amplified genes are PARK2, MECOM, RAD51B, THSD4, and SKAP1 (Table 2). Some of the prominently altered genes display gene-specific CNA in several cancer types, but often fail to meet the significance threshold.

It is unclear whether these altered genes drive the chromoanagenesis and tumorigenesis processes forward, or simply accompany them. The different chromoanagenesis processes are less likely to alter the copy number of a single gene, and are more likely to affect a chromosomal region. Nevertheless, the chromoanagenesis process might abrupt fragile sites (i.e., chromosomal regions with increased frequency of breaks). Previous studies have identified some of the altered genes as fragile sites: DMD, WWOX, PARK2, and LRP1B [22,23,24]. Other altered genes include known oncogenes and tumor suppressors: MECOM, RAD51B, ESR1, and also LRP1B (based on the COSMIC catalog gene census [25]).

### 3.5. Chromoanagenesis CNA Pattern Overlaps with Existing Knowledge

Many of the described tumor-specific CNA were previously detected and characterized in tumorigenesis studies. However, some of these CNA patterns were analyzed prior to the depiction of chromoanagenesis, and were not considered associated with the phenomenon. In BLCA, one of the four significantly amplified regions for chromoanagenesis is 6p22. There are four consecutive genes which pass the significance threshold, the most significant being E2F3 with a *p*-value of $7.9\times 10^{-9}$. E2F3 is a transcription factor that interacts directly with the retinoblastoma protein (RB1) to regulate the expression of genes involved in the cell cycle. The amplification of this region, and specifically E2F3 in bladder cancer, was associated with tumor cell proliferation [26]. The other three amplified regions in chromoanagenesis BLCA were also previously linked to bladder cancer; 1q23 [27], 3p25 [28], and 8q22 [29].

BRCA chromoanagenesis samples have three deleted regions, deleted 17q21 includes the oncogene BRCA1. Lettesier et al. [30] analyzed samples of breast cancer with copy number amplifications in 8p12, 8q24, 11q13, 12p13, 17q12, and 20q13. We found that amplification in 4 of those 6 chromosomal regions is also significantly associated with chromoanagenesis. The gene CSMD1, frequently altered in chromoanagenesis, is a known breast cancer tumor suppressor, associated with high tumor grade and poor survival [31,32].

GBM chromoanagenesis has a small amplification of three consecutive genes in 12q15, including the gene MDM2. MDM2 is transcriptionally regulated by p53. It promotes tumor formation by targeting p53 protein for degradation. Overexpression or amplification of this locus is detected in a variety of different cancers. Amplification of MDM2 without TP53 mutations was observed in gliomas [33,34]; this matches our observation, as GBM chromoanagenesis is not enriched for classic chromoanagenesis signature of TP53. Similarly, the CNA at 12q15 that includes MDM2 is associated with alteration in SARC [35].

### 3.6. Somatic SNV Reveal Chromoanagenesis Gene Differentiation

We further analyzed somatic SNV in chromoanagenesis samples for each of the 20 examined cancer types (see Methods). We tested the total number of somatic exome SNV, the number of affected genes, how many occurrences of loss-of-function (LOF), missense and synonymous mutations occurred, as well as the number of affected driver genes (based on the COSMIC catalog gene census [25]). For the most part, the total number of somatic SNV was mostly similar between chromoanagenesis and non-chromoanagenesis samples. None of the groups had exceedingly more SNV across all cancer types. A notable exception was UCEC, in which the non-chromoanagenesis samples had at least five times more SNV in all measured aspects. Aggregated SNV level-effects in chromoanagenesis are available in Appendix A.

For each gene, in each cancer type, we calculated separately the number of individuals with LOF, missense, non-synonymous (either LOF or missense), and synonymous mutations, in chromoanagenesis and non-chromoanagenesis samples. We tested the differences for each gene and each mutation type using the conservative Fisher’s exact test. Detailed results for genes with *p*-value smaller than $5\times 10^{-3}$ are available in Appendix A. This comparison enabled us to identify cancer driver genes related to chromoanagenesis and also, driver genes that specify non-chromoanagenesis tumorigenesis. The rate of the synonymous mutations for a specific gene can be considered as the mutation rate background.

The top four genes detected as likely chromoanagenesis inducing genes are TP53, ATRX and to a lesser extent: PPP2R1A and ST6GAL2. TP53 and ATRX are two prominent, known chromoanagenesis causing genes [36,37,38]. In 10 out of the 20 tested cancers (BLCA, BRCA, COAD, HNSC, LGG, LUAD, PAAD, PRAD, STAD, and UCEC), there were significantly more TP53 LOF or missense mutations in chromoanagenesis than in non-chromoanagenesis. In UCEC, 79.4% of chromoanagenesis classified samples had either a missense or LOF mutation in TP53, in comparison to only 17% in the non-chromoanagenesis samples (*p*-value: $1.22\times 10^{-36}$). ATRX had significantly more missense or LOF in chromoanagenesis samples in both LGG and SARC. In LGG, 53% of chromoanagenesis samples were mutated while only 27.1% of non-chromoanagenesis samples were mutated (*p*-value: $5.32\times 10^{-7}$). ATRX inactivation was linked to TP53 mutations and altered telomeres [39,40]. PPP2R1A was significantly more mutated in chromoanagenesis in UCEC (*p*-value: $1.33\times 10^{-5}$), and ST6GAL2 in LUAD (*p*-value: $1.12\times 10^{-6}$).

Many prominent oncogenes are SNV impaired at a higher rate in non-chromoanagenesis samples. The most substantial non-chromoanagenesis genes are PTEN, CIC, CASP8, KMT2D, ARID1A, RNF213, and PIK3CA. PTEN, an established tumor suppressor [41], is associated with many cancer types. In UCEC, the gene has missense or LOF in 72.5% of the non-chromoanagenesis samples, and in only 18.25% of the chromoanagenesis samples (*p*-value: $6.02\times 10^{-27}$). CIC has more damaging mutations in non-chromoanagenesis samples in STAD, LGG, COAD, and UCEC. In LGG, it is damaged in 25.2% of non-chromoanagenesis samples and is not damaged at all in chromoanagenesis samples (*p*-value: $3.65\times 10^{-13}$). KMT2D, ARID1A, RNF213 and PIK3CA present similar trends in both UCEC and STAD. CASP8 is commonly mutated in HNSC non-chromoanagenesis samples (*p*-value: $6.91\times 10^{-6}$).

### 3.7. Mutual Exclusivity Imply Distinct Tumorigenesis Pathways

Some of the examined cancer types include both genes frequently impaired (i.e., accumulated missense or LOF mutations) in chromoanagenesis samples, and genes frequently impaired in non-chromoanagenesis samples. We performed a mutual exclusivity analysis for the differentially impaired genes in each cancer type using cBioPortal [42,43]. The analysis tests whether we see less simultaneous mutations occur in a gene pair in the same patients than is expected by chance. We included several different research cohorts for each cancer type, derived from both TCGA and a number of additional resources. Only genes with mutual exclusivity *q*-value of <0.005 are presented. TP53, a top chromoanagenesis gene (and ATRX in LGG) is mutually exclusive from other cancer driver genes (Figure 6). Recurring genes in the non-chromoanagenesis samples include CIC, KMT2D, ARID1A, and RPL22.

Many of these mutually exclusive relationships were previously detected and studied. In LGG, the genes TP53 and ATRX are impaired in chromoanagenesis samples, while CIC and FUBP1 are impaired in non-chromoanagenesis samples, these mutually exclusive genes were connected to specific pathological and clinical characteristics [39]. In HNSC, the genes TP53 and HRAS impairment are mutually exclusive. Specifically, individuals with TP53 mutated HNSC have reduced immune activity while individuals with HRAS mutated HNSC have an increased immune activity [44]. In BLCA, mutations in FGFR3 (mutually exclusive to TP53) are correlated with bladder tumors of lower grade and stage [45].

Among the genes mutually exclusive with TP53 are several members of the ARID family (i.e., ARID1A, ARID1B, and ARID5B. Figure 6). The human ARID family contains 15 coding genes whose main function is in cell differentiation and proliferation, specifically in cancer-related signaling pathways. Mutations in ARID family members are common in many tumor tissues, and it is a sensitive marker for cancer prognosis or therapeutic outcome [46]. It was observed that mutations in ARID1A and TP53 are typically mutually exclusive in epithelial ovarian cancer [47]. In many gynecological cancers, the lack of ARID1A predicts early recurrence. Moreover, somatic ARID1A in these cancer types consist mostly of frameshift or nonsense mutations leading to LOF. It is likely that the mutual exclusivity between ARID1A and TP53 is explained by epigenetic signaling in gynecological cancers [48]. A proposed mechanism underlying the mutual exclusivity suggests that mutations in ARID1A contribute to the inactivation of p53-induced apoptosis. In healthy tissues, ARID1A suppresses the expression of the HDAC6 gene. However, in cancer samples with LOF of ARID1A, HDAC6 is elevated which in turn, represses apoptotic function of p53 [47].

The differences in SNV impaired genes across chromoanagenesis states is likely to imply on two distinct pathways in cancer development. A TP53-chromoanagenesis pathway, driven by DNA instability and DNA breaks, and a more diverse, cancer-type dependent, non-chromoanagenesis pathway that cover multiple processes as depicted by the major cancer hallmarks.

### 3.8. Chromoanagenesis Samples Are Mostly Not Signified by Distinct Clinical Characteristics

We compared all available clinical attributes between chromoanagenesis and non-chromoanagenesis samples for the 20 types of cancer. The analysis included demographic characteristics such as age, gender, race and ethnicity, tumor specific characteristics such as morphology, prior treatment, and tumor stage. Exposure features, such as BMI, smoking, and alcohol use history were also examined. In addition, we performed Cox-regression analysis for the 20 cancer types (Appendix A). For the most part, there were no distinct differences in the many variables tested between chromoanagenesis and non-chromoanagenesis samples. There were also no prominent differential survival trends favoring either chromoanagenesis or non-chromoanagenesis samples. These results are available in Appendix A.

Notably, there were three cancer types with varied morphology distribution between chromoanagenesis and non-chromoanagenesis samples: ESCA, SARC, and UCEC (Figure 7). In UCEC, 90.1% of the non-chromoanagenesis patients had endometrioid carcinoma, while 61.9% of the chromoanagenesis patients had serous cystadenocarcinoma. The distribution in morphology matches the molecular subtypes distribution [49,50,51] for these three cancer types. Interestingly, the integrated genomic characterizations for ESCA, SARC, and UCEC highlights genes detected in this study as chromoanagenesis-related genes, such as TP53, ATRX, PPP2R1A, and MDM2.

### 3.9. Chromoanagenesis Does Not Correlate with HPV

It was postulated that human papillomavirus (HPV) causes certain chromoanagenesis effects in infected individuals [3]. We collected HPV status for HNSC samples [9], and tested whether there is an enrichment for HPV infections within our classified chromoanagenesis samples. Out of the 171 non-chromoanagenesis samples, 13 (7.6%) were positive for HPV, and 7 out of the 63 chromoanagenesis samples (11.1%) were positive for HPV. These results suggest that HPV does not seem to induce chromoanagenesis-like patterns during tumorigenesis.

## 4. Discussion

We performed CNA, somatic SNV, and clinical data chromoanagenesis analyses for 20 cancer types. Chromoanagenesis samples presented distinct CNA patterns, mostly cancer-type specific. The somatic SNV analysis, however, revealed similar genic phenoms. Many of the observed CNA and somatic SNV patterns were previously independently reported, but some were not associated with chromoanagenesis. We offer these reported patterns as further evidence to the validity of our methodology and discoveries and suggest chromoanagenesis as a possible driving force for known oncogenic CNA phenoms. Providing this additional context can aid in better defining subtypes of cancer, as well as revealing underlying shared tumorigenesis mechanisms. Surprisingly, we hardly found any distinguishing clinical features between the proposed tumorigenesis routes, despite existing reports on a diminished survival rate in chromoanagenesis [37,52]. It is still possible that the different subtypes of chromoanagenesis underlie the mostly homogeneous results. In this case, there might be clinical properties obscured by considering all individuals with chromoanagenesis as a unified group. In this study, we limited the analysis to genic CNA; however, a more general non-gene-centric CNA predictor could better distinguish the different chromoanagenesis subtypes, and provide additional insights regarding overlooked non-genic CNA regions. Notably, TCGA data only provides a single time point for each tumor sample and therefore testing chromoanagenesis intra-tumor heterogeneity is not possible [53,54].

TP53 is the most common gene damaged in many chromoanagenesis samples. TP53 has a much higher rate of missense or LOF mutations in chromoanagenesis [37], while some other known driver genes are often damaged in non-chromoanagenesis individuals. There is a pattern of mutual exclusivity between genes damaged in chromoanagenesis and non-chromoanagenesis samples. As some types of chromoanagenesis are considered to occur in an early stage of tumorigenesis [7], it is possible that there are two main distinct pathways in the observed samples: one driven by a single dramatic chromosomal rearrangement event and the other process relies on accumulated point mutations in crucial cancer genes. Each process is propelled by its own driver genes and a distinct primary tumorigenesis process.

The high frequency of the chromoanagenesis phenomenon in cancer became evident in recent years [7]. It was also detected as a possible cause for other serious conditions, such as congenital disorders [55,56,57]. As chromoanagenesis was only defined with the advances in technologies in recent years, the extent of the phenomenon is widely unknown. Quite surprisingly, several cases of chromoanagenesis were reported from the germline of healthy individuals [58]. Chromoanagenesis in healthy individuals mostly generates copy number neutral translocations and inversions. These healthy individuals often suffer natural abortions or give birth to offspring with congenital disorders [59]. Evidently, the consequences of a chromoanagenesis event are dependent on the nature of the chromosomal rearrangements it induces. The abundance and variety of cancerous chromoanagenesis samples provides an ideal resource to investigate the chromoanagenesis phenomenon, which is probably understudied in the non-cancerous context.

## Figures and Tables

**Figure 1 cancers-13-04197-f001:**
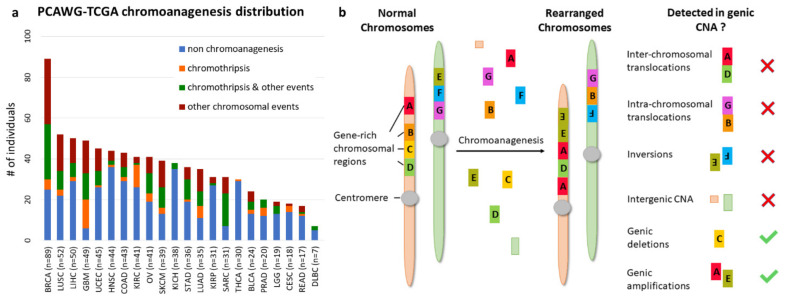
Chromoanagenesis in PCAWG–TCGA joint samples. The PCAWG cohort includes chromoanagenesis status for 799 individuals with 22 types of cancer. Notably, TCGA samples normally only have CNA data rather than whole genome sequencing. (**a**) Cancer type and chromoanagenesis subtypes distribution for the 799 individuals. (**b**) A schematic depicting the common chromosomal abnormalities caused by chromoanagenesis, and presents which of the abnormalities can be captured when considering only genic CNA data.

**Figure 2 cancers-13-04197-f002:**
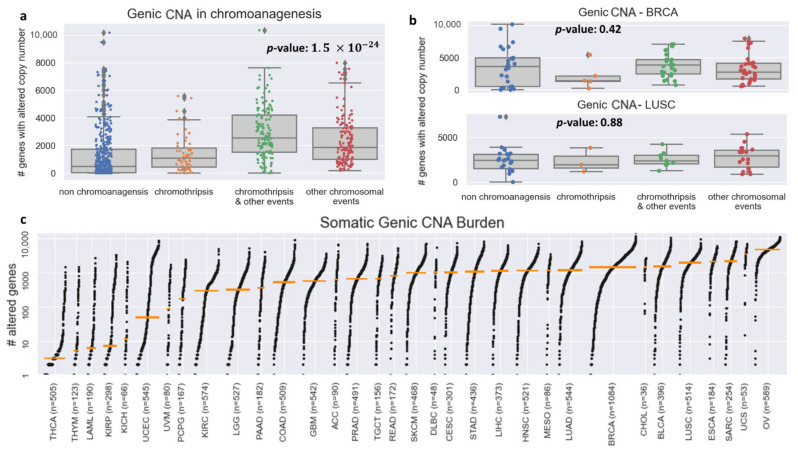
Genic CNA frequency in PCAWG–TCGA. (**a**) Boxplots for the number of genes with altered copy number for each chromoanagenesis subtype. (**b**) Boxplots for the number of genes with altered copy number for BRCA and LUSC patients. (**c**) Genic CNA number distribution in each of the 33 types of cancer included in TCGA. The cancer types are sorted by the median number of copy-altered genes (marked by an orange bar).

**Figure 3 cancers-13-04197-f003:**
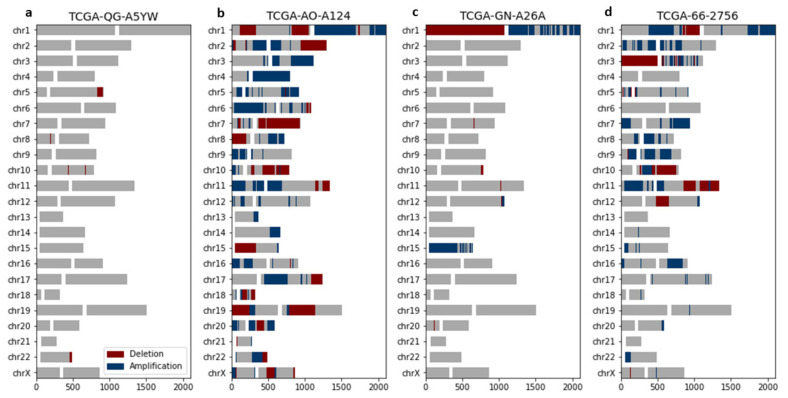
Chromoanagenesis samples present chromosome-specific CNA oscillations. CNA patterns for four representative PCAWG–TCGA samples. For each chromosome, gene amplification and gene deletion are depicted by blue and red, respectively. Centromeres are signified by a white gap, separating the *p*-arm (left) from the q-arm (right). (**a**) TCGA-QG-A5YW, a COAD patient without chromoanagenesis with a total of 7 oscillations. (**b**) TCGA-AO-A124, a BRCA patient without chromoanagenesis with a total of 118 oscillations. (**c**) TCGA-GN-A26A, a SKCM patient with chromoanagenesis with a total of 35 oscillations (primarily in chromosomes 1 and 15). (**d**) TCGA-66-2756, a LUSC patient with chromoanagenesis with a total of 122 oscillations (primarily in chromosome 3).

**Figure 4 cancers-13-04197-f004:**
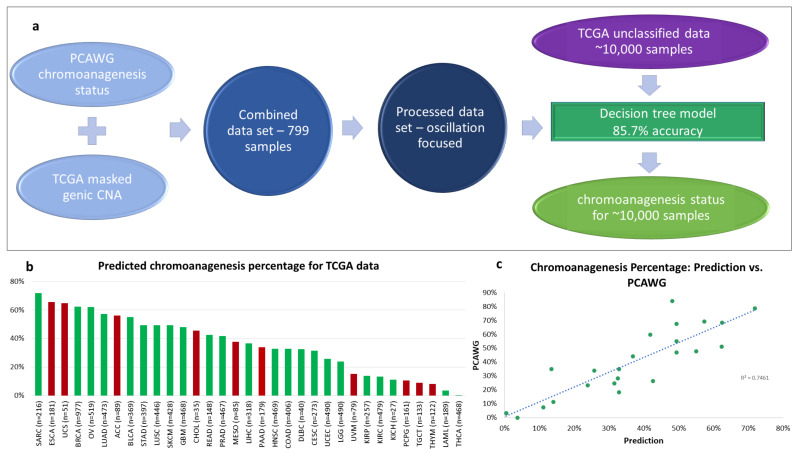
Chromoanagenesis prediction. PCAWG and TCGA data were integrated and processed to train a machine learning (ML) model (**a**) The selected model has an accuracy rate of 85.7%. It was applied to predict chromoanagenesis status for the remaining 9929 TCGA individuals. (**b**) Predicted chromoanagenesis rate for all 33 cancer types in TCGA. Bars of the histogram representing chromoanagenesis percentage estimates for cancer types not included in the training set are colored red. (**c**) Correlation between predicted chromoanagenesis rate and the PCAWG reported chromoanagenesis rate for shared cancer types.

**Figure 5 cancers-13-04197-f005:**
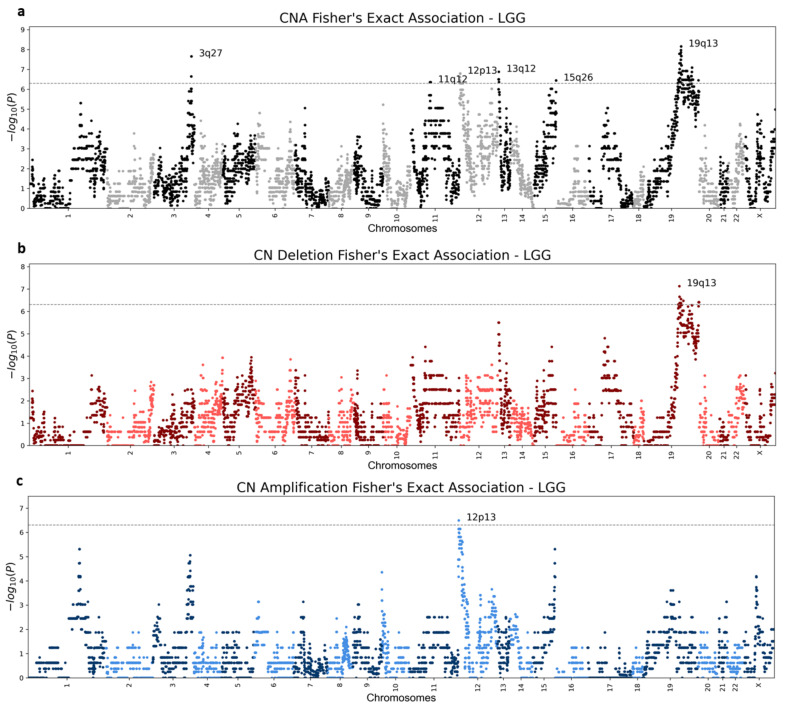
LGG (Brain lower grade glioma) Manhattan plots. Genic Manhattan plot over Fisher’s exact test *p*-values between LGG chromoanagenesis samples and non-chromoanagenesis samples. (**a**) LGG Manhattan CNA (combined deletion or amplification) plot. (**b**) LGG Manhattan plot of deletion events. (**c**) LGG Manhattan plot for amplification events. The sequential chromosomes are colored differently for visualization purposes. The conservative significance statistical threshold is set to $5\times 10^{-7}$.

**Figure 6 cancers-13-04197-f006:**
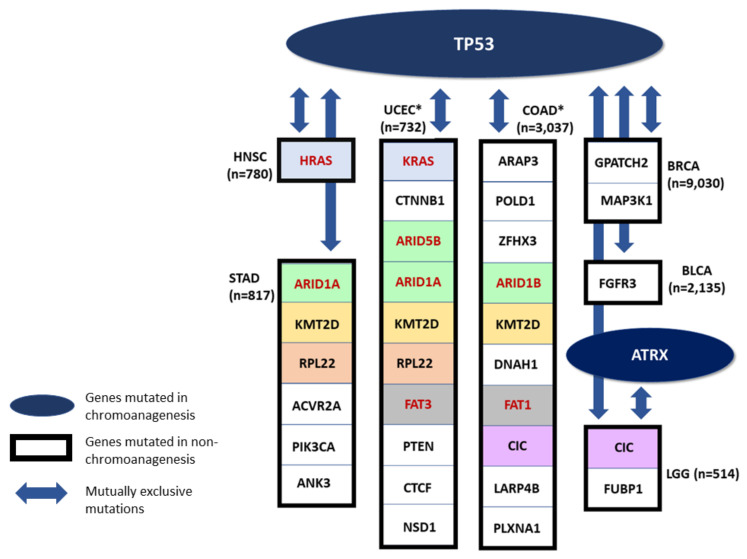
Mutually Exclusive Genes. A schematic presenting mutual exclusivity analysis for chromoanagenesis differentially impaired genes. TP53 and ATRX (in LGG) are significantly more impaired in chromoanagenesis, and are also mutually exclusive from genes significantly more impaired in non-chromoanagenesis individuals. Only genes with mutual exclusivity *q*-value < 0.005 are shown. Genes that appear in more than one cancer type are indicated by the same background color. Paralogous genes are marked with red font and colored with a similar background. * Only the top 10 genes are shown, for cancer types with more differentially-impaired, mutually-exclusive genes.

**Figure 7 cancers-13-04197-f007:**
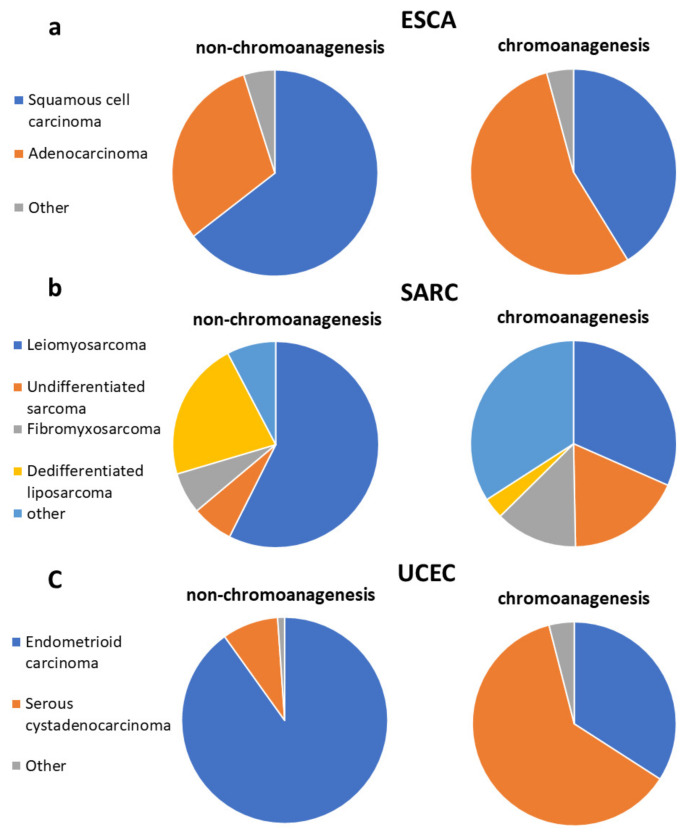
Chromoanagenesis morphological distribution. Pie charts representing the morphology distribution in chromoanagenesis and non-chromoanagenesis samples for (**a**) ESCA, (**b**) SARC, and (**c**) UCEC.

**Table 1 cancers-13-04197-t001:** Number of significant CNA regions per cancer type.

Cancer Type		# Amplified Regions	# Deleted Regions	# Additional Altered Regions
BLCA	Bladder urothelial carcinoma	4		3
BRCA	Breast invasive carcinoma	20	3	14
CESC	Cervical squamous cell carcinoma and endocervical adenocarcinoma			2
COAD	Colon adenocarcinoma	1	1	
GBM	Glioblastoma multiforme	1		1
HNSC	Head and neck squamous cell carcinoma			5
LGG	Brain lower grade glioma		1	5
LUAD	Lung adenocarcinoma	1		6
LUSC	Lung squamous cell carcinoma	1		
OV	Ovarian serous cystadenocarcinoma			2
PRAD	Prostate adenocarcinoma			1
SARC	Sarcoma			1
SKCM	Skin cutaneous melanoma	3		3
STAD	Stomach adenocarcinoma	16	4	38
UCEC	Uterine corpus endometrial carcinoma	166	57	74

**Table 2 cancers-13-04197-t002:** Significant CNA genes.

Gene	Gene Full Name	Amplified in	Deleted in	Altered in	Is Driver
ANKS1B	Ankyrin Repeat And Sterile Alpha Motif Domain Containing 1B	UCEC			−
CSMD1	CUB And Sushi Multiple Domains 1			BRCA	−
DLG2	Discs Large MAGUK Scaffold Protein 2	UCEC			−
DMD	Dystrophin	UCEC, ESCA *, STAD *			−
ELAVL1	ELAV Like RNA Binding Protein 1			UCEC	−
ESR1	Estrogen Receptor 1			UCEC	+
FGF14	Fibroblast Growth Factor 14		PRAD		−
KAZN	Kazrin, Periplakin Interacting Protein			UCEC	−
LRP1B	LDL Receptor Related Protein 1B	UCEC, OV *			+
LSAMP	Limbic System Associated Membrane Protein			UCEC, STAD *	−
MACROD2	Mono-ADP Ribosylhydrolase 2		STAD		−
MECOM	MDS1 And EVI1 Complex Locus		UCEC		+
PARK2	Parkin RBR E3 Ubiquitin Protein Ligase		COAD		−
PDE4D	Phosphodiesterase 4D	STAD, UCEC, ESCA *		BLCA	−
PGM5	Phosphoglucomutase-Related Protein			UCEC	−
RAD51B	RAD51 Paralog B		UCEC		+
SKAP1	Src Kinase Associated Phosphoprotein 1		UCEC		-
THSD4	Thrombospondin Type 1 Domain Containing 4		UCEC		−
WWOX	WW Domain Containing Oxidoreductase	UCEC			−
ZMAT4	Zinc Finger Matrin-Type 4				−

* Prominently altered but fail to meet the significance threshold.

## Data Availability

The data supporting the findings of this study are publicly available in the PCAWG and GDC cancer portal.

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
