# Peer review of "Chromoanagenesis Landscape in 10,000 TCGA Patients"

_cancers, 2021, doi:10.3390/cancers13164197_

Round 1

Reviewer 1 Report

Thanks for the clarifications. They have responded satisfactorily to my questions and have completed the paper according to my recommendations.

Author Response

Thank you.

Reviewer 2 Report

The authors have addressed all points raised by this reviewer. As pointed out in the first revision, the authors could have chosen a non gene-centric method to analyze CNAs and compare them against each other as a predictor of chromoanagenesis. A statement in this direction could be included in the Discussion. Nonetheless, small clarifications inserted in the revised text make the overall message of the manuscript stronger than in the previous version.  Although this reviewer still considers the Results section too long (and thus it could be sharper), I can also understand the point of view from the authors to include all aspects of their findings. All in one, the manuscript is suitable for publication upon minor changes.

Author Response

This manuscript is a resubmission of an earlier submission. The following is a list of the peer review reports and author responses from that submission.

Round 1

Reviewer 1 Report

In the present manuscript, Rasnic and Linial attempted to describe chromoanagenesis events in large scale sample sets using PCAWG and TCGA data. Chromoanagenesis, which includes events such as chromothripsis, chromoplexy and chromoanasynthesis, are complex genomic events with little known about the mechanism by which they happen and, most importantly, how are maintained in the cell fate. Therefore, studies to describe the genomic background on samples carrying chromoanagenesis are of utmost interest. The authors intended to correlate these events with genomic features, mainly copy number alterations (CNA) and single nucleotide variants (SNV), from different points of view including a machine learning model. To this reviewer, the manuscript should be more focused and detailed on the data themself rather in justifying their findings. Moreover, some of the approaches and interpretations the authors have considered are limited and do not reach solid conclusions. Major concerns are:

1- Description of chromothripsis (described within the term chromoanagenesis in the mansucript) using PCAWG data has been previously reported in a recent article by Cortés-Ciriano et al (PMID: 32025003). The authors should explain how they identified chromoanagenesis events in their study and compare their approach with those already reported. This might explain differences in the frequency of samples affected by these events.

2- The integration of chromoanagenesis events and copy number alterations results intriguing. Nevertheless, the gene-centered approach is misleading because CNAs are considered to affect a segment of the genome. In addition, the correlation between frequency of CNAs and chromoanagenesis events might be due to the high levels of chromosome instability, which is a hallmark of the cancer cells. How do the authors integrate this phenomenon into their analysis?

3- The analysis of altered genes per tumor type reflects the levels of CNA per tumor. This information has been extensively established in other studies. In a particular example the authors provided (Figure 1a), it is necessary to establish what are the "other events" that seem to be guiding genes with CN, which most likely will correspond to gains and losses typically observed in tumors. These data suggest that CN affecting individual genes are independent from chromothripsis events.

4- Figure 3 is rather misleading. The data do not proof that CNA specific events are the consequence of chromoanagenesis. The section "predicting chromoanagenesis at high accuracy" is mostly descriptive and inaccurate.

5- The sentence "The chromoanagenesis process is likely to abrupt fragile sites" is too speculative based on the data generated. It is well known that focal deletions occur preferentially at sites of fragile sites, although its causes are still unknown.

6- I encourage the authors to strengthen their analyses using clinical features. At least for chromothripsis, these events have been associated with aggressive forms of cancer and poor patient survival (reviewed in Forment JV et al., Nat Rev Cancer. 2012 and Kloosterman WP et al., Curr Opin Oncol. 2014).

Minor concerns:

1- The authors are sometimes using the term CNV instead of CNA (eg, Figure 4a).

2- The Results section includes quite some text that is more suggestive of Discussion than Results. Please revise the distribution of these sections.

3- The title might be too broad and uncertain.

Reviewer 2 Report

Comment:

Building a machine learning algorithm for cancer detection inspired and conjured the readers up hope to novel cancer detections and new comprehensive views on cancer treatment. However, conventional ML models did not break down old modes of known cancer biomarkers from the same data sets - neither light up a new way nor lead to a novel theory. Like ML training, the readers need to be trained to know it – this manuscript contributes to such movement. Some specifics below should be incorporated for clarity.

Specific comments:

  • Page 1, Lines 10-11: “The Pan-Cancer Analysis of Whole Genome (PCAWG) 10 to build a machine learning algorithm that detects chromoanagenesis with high accuracy.” How did they define “accuracy”? They used ~10,000 TCGA samples and PCAWG to build the ML algorithm and verify it with the same data set (Lines 21-23).  Some external data sets (their own clinical samples) should be used to verify it, not with the same data set for ML model selection and training (Line 75).  Lines 196-197: “The machine learning model was applied on all remaining 9,929 TCGA samples from 196 all 33 cancer types reported in TCGA (Figure 4a, Table S1).” What’s the reason to divide which part was for ML modeling and which part was for verification?
  • Line 24: “high accuracy (86%)” – How much did the pathologists get in clinics?
  • Page 2, Lines 46-53: “The most exhaustive research on chromoanagenesis was performed as a part of The Pan-Cancer Analysis of Whole Genomes (PCAWG) study[5]. The analysis included 2,658 cancer genomes and their matching normal tissues across 38 tumor types. The study confirmed that chromoanagenesis is common in many cancer types. There is an overlap of 799 of the examined genomes (from 22 tumor types) with The Cancer Genome Atlas (TCGA) project. We utilized this overlap in order to curate a data set with TCGA genic CNA data, and PCAWG chromoanagenesis labeling. We utilized the data set to create a highly accurate machine learning model that identifies chromoanagenesis and employed it on ~10,000 cancerous samples from TCGA.” Some fine-tone might help the readers to gain traction of the logic.
  • “Many of the found somatic single nucleo-55 tide variants (SNV) and CNA patterns match previous studies, as we concur on chromoanagenesis 56 related genes.” (Lines 55-57). The same data set should be used for self-assurance, as shown in Line 66 given “PCAWG thoroughly described each individual chromosomal state” (Fig 1).
  • The font of Lines 108-112 differed from the font of Lines 113-117. Any meaning?
  • Lines 189-190: “We trained a decision tree on 85% of the data to identify whether a sample has chromoanagenesis (see Methods)” – why did they pick up 85%? Any clinical reasoning to support such a number?
  • How did the justify “We reached an 85.7% accuracy rate, with 88.9% accuracy, 190 83.6% specificity, and 78.4% sensitivity for having chromoanagenesis” sufficient to call the shot in clinics?
  • Fig 6 - TP53 and ATRX. What happened to EGFR or other biomarkers of cancer drivers?
  • Lines 436: “TP53, ATRX, PPP2R1A and MDM2.” – Nothing is novel finding with this ML algorithm but a confirmatory set.
  • In discussion, the shortcoming of ATCG data sets (doi: 10.1186/s12935-014-0115-7) should be pointed out as single-cell transcriptomes came to change the picture (doi: 10.1093/carcin/bgy052). Neither mentioned any therapeutics nor much tighten in clinical outcomes from this data science approach led to new hypotheses.

Reviewer 3 Report

Interesting article, but I think it should be completed by relating the results to clinical or therapeutic implications that justify the interest in characterizing the chromoanagenesis in the samples.

Line 12: explain the meaning of TCGA

In the introduction, the three sub-types of chromoanagenesis should be explained in more detail, better explaining the characteristics and differences of each of them.

Line 110: when the results are explained, only cases of chromothripsis are discussed, no cases of the other subtypes are detected or the other cases encompass some of the subtypes. How often are there references to each type? Are the results found representative?

Line 112: Can the only other, non-chromothripsis, complex chromosomal events samples be considered chromoanagenesis samples?

When you talk about chromoanagenesis being detected in healthy people, is that related to a higher chance of developing a tumor?

When reading the article I am not clear about the implications of detecting chromoanagenesis in tumors, does it serve to make a prognosis? If it can appear in healthy people, is it a cause of the appearance of a tumor? In the article it is said that chromoanagenesis-TP53 path-413 way, driven by DNA instability and DNA breaks, therefore, would it be a good prognostic factor?

Reviewer 4 Report

It is indicated that chromanogenesis is observed in healthy normal individuals. Some discussion of these findings should be included to clarify the clinical significance of the findings in cancer patients..

How are these identified. Are specific chromosomal patterns inserted, or is this a designated/described abnormality. A more detailed description of chromogenesis would be helpful.

Lines 102 It is unclear how it is determined whether a given specimen has chromotripsis, other complex chromosomal events, or both. What events are required to make the diagnosis of chromotripsis. The features which were analyzed are described in lines 83-87, how are these selected to designate one of the 3 groups? This is never adequately described, especially if you are doing machine learning to analyzes samples.

Line 145. The potential degree of heterogeneity across these samples is considerable, with differences according to the presence/absence of chromanogenesis, differences according to chromotripsis or complex chromosomal abnormalities within chromanogenesis, and then differences according to CNA, genes involved, and SNV within the latter groups. A comment on this heterogeneity would be helpful. With regard to the samples from the TCGA, are these a single biopsy from each tumor, which would not address heterogeneity within the tumor.

It is stated in line 174 that some features were engineered to express the difference between maximal number of oscillations (in a chromosomal arm) and the average number of oscillations (across all chromosomal arms). How are differences between chromosomes integrated into a definition of chromotripsis vs other complex chromosomal abnormalities?

Line 202: we observed that the chromoanagenesis rate is very similar for most cancer types, with ?2 of 0.7461. What exactly is meant by this observation? If the rate of development of chromoanagenesis over time is the same for different cancers, then why is the incidence so different between cancers?

It is stated in line 273 that “Some prominent and significant CNA consist only of a single gene.” One must ask, given all of the other CNAs in these tumors, what is the significance of this finding, unless these are prominent oncogenes.

It is stated in Line 327, the total number of somatic SNV was mostly similar between chromoanagenesis and non-chromoanagenesis samples. None of the groups had exceedingly more SNV across all cancer types. Whereas, in Line 412, the differences in SNV impaired genes across chromoanagenesis states is likely to imply on two distinct pathways in cancer development. A chromoanagenesis-TP53 pathway, driven by DNA instability and DNA breaks, and a more diverse, cancer-type dependent, non-chromoanagenesis pathway that cover multiple processes as depicted by the major cancer hallmarks. Are the authors suggesting that the only differences in the pathways is with regard to the SNV impaired genes.
